# Circular RNAs: Emerging Regulators of the Major Signaling Pathways Involved in Cancer Progression

**DOI:** 10.3390/cancers13112744

**Published:** 2021-06-01

**Authors:** Maria Papatsirou, Pinelopi I. Artemaki, Paraskevi Karousi, Andreas Scorilas, Christos K. Kontos

**Affiliations:** Department of Biochemistry and Molecular Biology, Faculty of Biology, National and Kapodistrian University of Athens, 15701 Athens, Greece; papatsir@biol.uoa.gr (M.P.); partemaki@biol.uoa.gr (P.I.A.); pkarousi@biol.uoa.gr (P.K.); ascorilas@biol.uoa.gr (A.S.)

**Keywords:** circRNA, miRNAs, signal transduction, WNT/β-Catenin, PI3K/AKT, MAPK/ERK, metastasis, biomarkers, targeted therapy, signaling cascades

## Abstract

**Simple Summary:**

Circular RNAs (circRNAs) are single-stranded RNA molecules that form a covalently closed loop structure. They are characterized by distinct features and multifariously implicated in the regulation of both physiological and pathological states. The aberrant expression of circRNAs has been evidenced in various malignancies. Circular transcripts can effectively modulate gene expression. The most prevalent manner through which circRNAs promote cancer development and progression is their interaction with key components of major signaling pathways. In particular, abnormally expressed circRNAs can dictate the crosstalk between signaling cascades. In recent years, there has been great progress regarding circRNA research in the context of cancer progression, and various regulatory axes have been described. As our knowledge of signaling regulation by circRNAs continuously expands, novel therapeutic approaches can be assessed and established, seeking to overcome clinical challenges, such as the treatment of cancer patients with distant metastasis and those who relapse.

**Abstract:**

Signal transduction is an essential process that regulates and coordinates fundamental cellular processes, such as development, immunity, energy metabolism, and apoptosis. Through signaling, cells are capable of perceiving their environment and adjusting to changes, and most signaling cascades ultimately lead to alterations in gene expression. Circular RNAs (circRNAs) constitute an emerging type of endogenous transcripts with regulatory roles and unique properties. They are stable and expressed in a tissue-, cell-, and developmental stage-specific manner, while they are involved in the pathogenesis of several diseases, including cancer. Aberrantly expressed circRNAs can mediate cancer progression through regulation of the activity of major signaling cascades, such as the VEGF, WNT/β-catenin, MAPK, PI3K/AKT, and Notch signaling pathways, as well as by interfering with signaling crosstalk. Deregulated signaling can then function to induce angiogenesis, promote invasion, migration, and metastasis, and, generally, modulate the hallmarks of cancer. In this review article, we summarize the most recently described and intriguing cases of circRNA-mediated signaling regulation that are involved in cancer progression, and discuss the biomarker potential of circRNAs, as well as future therapeutic applications.

## 1. Introduction

Cell signaling can be defined as the fundamental process of extra- and intercellular communication, which controls basic cellular activities that are essential for survival. The ability of cells to receive, process, and respond to signals from their environment is the foundation for their growth, division, differentiation, genetic stability, and cellular homeostasis [1,2]. There is a plethora of signaling pathways that dictate complex cellular responses, and while they can consist of varying components and have distinct effects, many of them intertwine and affect each other [3,4]. This crosstalk is observed in various tissues, wherein two or more signaling pathways reinforce each other.

Given that signal transduction is vital for maintaining normal cellular characteristics, the deregulation of signaling pathways is intricately involved in the development of pathological states, such as autoimmunity and cancer [5,6]. In particular, key components of various pathways can be mutated, causing the transition of proto-oncogenes to oncogenes and the hyperactivation of these pathways, while the downregulation of tumor inhibitors fosters cancer cell proliferation and metastasis [7]. For instance, the PI3K/AKT (AKT1) pathway is involved in the regulation of the cell cycle; however, when overactivated, it can lead to tumorigenesis, evasion of apoptosis, invasion, and a stem-like phenotype [8].

The critical role of signaling pathways is irrefutable in both cancer initiation and progression. In the last decades, the 5-year survival rate of patients with solid tumors and many hematological malignancies has increased, as a result of the clinical advances that allowed earlier detection and local tumor control [9,10]. However, cases of late relapses and late mortality have increased as well, rendering the invasive potential and the formation of early metastasis major challenges that need to be confronted [9,11]. The progression of cancer is a complex multistep process that is not sufficiently understood, and is associated with increased clinical aggressiveness; therefore, it is difficult to be prevented or controlled [12]. For these reasons, it is imperative to further explore the genetic mechanisms underlying cancer progression, and circular RNAs (circRNAs) have emerged as key molecules in this context that interact multifariously with major signaling pathways.

circRNAs constitute a type of evolutionarily conserved, single-stranded RNA molecules that form a covalently closed loop, and their continuous structure denotes the lack of exposed 5′ and 3′ ends and polyadenylation tails [13]. The researchers’ interest in circRNAs has peaked in recent years, due to advances in high-throughput strategies which have revealed their unique characteristics and great potential [14]. More specifically, circRNAs have a broad expression range in diverse forms of life and are resistant to degradation by exonucleolytic activity; hence they are generally more stable than linear RNAs. Moreover, circRNAs are widely expressed, both in the cytoplasm and nucleus, and several studies have evidenced a cell-, tissue- or developmental-stage–specific expression pattern [13,15].

Numerous evidence supports that circRNAs are correlated with the progression of human diseases through their interaction with signaling pathways, and their biomarker potential has been proved by multiple studies [16,17]. In particular, circRNAs are generally proven to intervene in tumor stemness, migration, invasion, and metastasis [18]; for instance, they are involved in the activation of the epithelial–mesenchymal transition (EMT) process via the WNT/β-catenin signaling pathway, affecting directly the tumor progression [19,20]. Taking into consideration such findings that continue to accumulate, this review aims to summarize current data regarding the role of circRNAs in the regulation of major signaling pathways involved in cancer progression.

## 2. Biogenesis of circRNAs

circRNAs are formed through the back-splicing of a primary transcript, when the 3′ end of an exon (donor site) is directly joined to the 5′ end of an upstream exon (acceptor site) (Figure 1) [21,22]. Several mechanisms for circRNA biogenesis have been proposed so far, the two most common being “lariat driven circularization” and “intron pairing-driven circularization”.

In the first case, which is also referred to as “exon-skipping”, circularization takes place during splicing, where circRNAs are formed from lariat intermediate structures [5]. Three different types of circular transcripts derive from these internal splicing reactions: exonic circRNAs (EcircRNAs) from which the flanking introns are removed, intronic circRNAs (ciRNAs) that consist exclusively of introns [23], and exon–intron circRNAs (EIcircRNAs), when exons and introns are retained in the final structure [24]. Meanwhile, “intron pairing-driven circularization” is a more direct mechanism for circRNA formation, through which EcircRNAs and EIcircRNAs are produced. This process is based on the proximity of splice sites that mediates the hybridization of flanking introns [5]. In several instances, the circularization is further facilitated by reverse complementary sequences within the flanking regions, such as ALU repeat elements that can also dictate alternative circularization by forming different ALU pairs [25].

**Figure 1 cancers-13-02744-f001:**
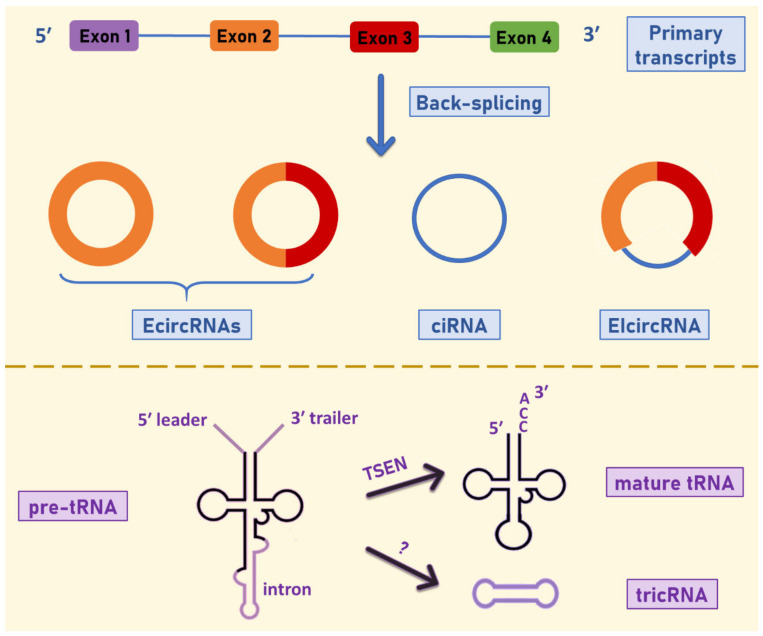
Biogenesis and types of circRNAs. Several types of circRNAs may derive from the back-splicing of a primary transcript. Exonic circRNAs (EcircRNAs) consist of one or more exons and intronic circRNAs (ciRNAs) are composed of a single intron. Moreover, an exonic–intronic circRNA (EIcircRNA) is produced, when both exons and introns are retained during circularization. A circRNA may also derive from the circularization of a pre-tRNA intron, when this latter is spliced out during the maturation process of a tRNA; such circRNAs are called tricRNAs. The tRNA-splicing endonuclease (TSEN) complex catalyzes the cleavage of an intron from an intron-containing pre-tRNA, whereas the enzyme that is responsible for the intron circularization has not been identified yet.

circRNA formation can also be mediated by RNA-binding proteins (RBPs), through their binding to flanking intronic sites which leads to hybridization. The most common examples include quaking (QKI) [26], muscleblind-like protein 1 (MBNL1) [21], and the RNA-editing enzyme ADAR [27]. These proteins can have a different effect and mechanism of action regarding circRNA formation; of note, QKI dimerization enhances circularization, while ADAR disrupts the pairing of flanking introns. Regarding the circRNAs deriving from pre-tRNA molecules (tricRNAs), the exact process for circularization is still unclear. The tRNA splicing endonuclease (TSEN) complex is involved in the mechanism and cleaves a pre-tRNA that contains an intron which is then ligated in order to form a circle [22].

## 3. Functions of circRNAs and Their Involvement in Signal Transduction

The main biological functions of circRNAs have been elucidated in both normal and pathological states (Figure 2). The most well-investigated mode of action is the miRNA sponging ability of circRNAs, where they act as competing endogenous RNAs (ceRNAs) [28]. Several circRNAs contain miRNA response elements (MREs) within their sequence, and, thus, can effectively sequester and prevent them from binding to their mRNA targets, ultimately affecting translation rates [29]. In detail, circRNAs can obstruct the complementary pairing of miRNAs to their target mRNAs 5′ or 3′ untranslated regions (UTRs), while they can also stabilize miRNAs, trigger their function, or protect them from degradation. The most well-known example of miRNA-sponges is CDR1as, a circRNA that contains more than 70 conserved binding sites for miR-7-5p and strongly affects its activity on downstream mRNA targets [29,30].

However, most circRNAs do not contain plenty of miRNA-binding sites and exert their regulatory function through other ways, such as protein sponging [21]. For example, circ-MBL (gene of origin: *MBNL1*) contains binding sites for the MBNL1 protein, evidencing that protein production and the circularization of transcripts from the same gene are closely related. Furthermore, circRNAs can decoy and stabilize proteins to a specific cellular compartment, affecting downstream targets. This process includes the binding and retaining of an RBP, which leads to higher expression rates of the protein target genes in this particular compartment [31]. An additional interaction of circRNAs and proteins is the ability of circRNAs to function as scaffolds for protein complexes, assisting their interaction. For instance, circ-AMOTL1 functions as a scaffold that facilitates the phosphorylation of AKT1 by PDK1 [32].

Albeit initially considered as non-coding RNAs having only a miRNA-sponging function, some circRNAs have been shown to contain multiple open reading frames (ORFs) and encode proteins, several of which are involved in major cellular processes [33]. The translation occurs in a Cap-independent manner and several circRNAs possess at least one internal ribosome entry site (IRES), while the process can be promoted by the nucleotide modification N^6^-methyladenosine (m^6^A) [33,34]. Additionally, circRNAs— particularly ciRNAs and EIcircRNAs which are mainly located in the nucleus—can regulate their parental gene transcription by interacting with the Polymerase II complex [23]. Finally, transcription and alternative splicing are also affected by circRNAs, considering that the circularization process is often competing with canonical splicing of the same pre-mRNA [21].

In several instances, the involvement of circRNAs in the deregulation of signal transduction has been evidenced, supporting the key role of circRNAs in cancer initiation, progression, metastasis, stemness, and resistance to therapy [35]. More specifically, circRNAs can regulate the cell cycle by exerting their sponging ability; for instance, circ-ZNF292 inhibits cell proliferation and promotes apoptosis of hepatocellular carcinoma cells by regulating the WNT/β-catenin pathway [36]. An additional example is circ_0005273 (gene of origin: *PTK2*), which regulates the Hippo/YAP (YAP1) pathway through sponging miR-200a-3p, in order to promote breast cancer tumorigenesis [37]. Proteins encoded by circRNAs are implicated in signaling regulation as well. For instance, recent data support that a novel protein encoded by circ-SMO is crucial for Hedgehog signaling and drives glioblastoma tumorigenesis [38]. Similarly, a protein encoded by circ-PPP1R12A activates the Hippo/YAP pathway and promotes the migration, invasion, and proliferation of colon cancer cells [39]. The interaction of circRNAs with major components of signaling cascades has been confirmed in numerous cases of progressive cancers.

## 4. CircRNA-Mediated Regulation of Major Signaling Pathways

### 4.1. VEGF

The vascular endothelial growth factor (VEGF) constitutes a family of signal proteins that are essential for the formation of new blood vessels and the growth of pre-existing ones. During this process, VEGF triggers gene expression, which, then, stimulates mitosis and migration of endothelial cells [40]. The induction of VEGF signaling occurs when VEGF ligands, such as VEGFA and VEGFC, bind to tyrosine kinase receptors (VEGFR1 (FLT1), VEGFR2 (KDR), and VEGFR3 (FLT4)) that are attached to the cell membrane. Following the receptors dimerization and *trans*-phosphorylation, they are activated, and subsequently, they activate downstream signaling pathways, for example, the RAS/MAPK, PI3K/AKT, and FAK (PTK2)/SRC cascades [41,42].

Solid tumors need sufficient and constant blood supply in order to obtain adequate oxygen and nutrients for growth and metastasis. For this reason, the overexpression of VEGF-mediated signaling through circRNAs, and the consequent promotion of angiogenesis and lymphangiogenesis is crucial for cancer cells, hence it has been the focus of several studies (Table 1). An exemplary case of circRNA-mediated signaling crosstalk in cancer is the effect of circ-MYLK on bladder cancer progression [43]. More specifically, circ-MYLK is upregulated in bladder cancer cells and tissues and its expression is positively correlated to the progression of tumor stage and grade. Mechanistically, it exerts an oncogenic role through binding miR-29a-3p and negating its restrictive regulation on VEGFA. The subsequent upregulation of VEGFA mediates the activation of the downstream RAS/RAF/MEK/ERK pathway, resulting in the promotion of cancer cell proliferation, angiogenesis, and EMT [43]. The same circRNA was also proved to regulate the function of VEGFC in renal cell carcinoma, by binding miR-513a-5p as a ceRNA [44]. This interaction leads to enhanced invasive and migratory capabilities of renal tumor cells, as well as larger tumor size and distant metastasis.

The interplay between the VEGF and PI3K/AKT pathways via circRNAs has been reported as well. circ-NFIB_1_ (gene of origin: NFIB) has an anti-tumor effect in pancreatic cancer, through binding miR-486-5p, thus restoring the expression levels of its target, *PIK3R1*, which encodes the regulatory subunit 1 of PI3K [45]. PIK3R1 then reduces the expression levels of VEGFC through the obstruction of the PI3K/AKT pathway, ultimately suppressing migration, lymphangiogenesis and lymphatic metastasis. Moreover, circ_0023404 (gene of origin: *RNF121*) promotes metastasis and chemoresistance of cervical cancer and obstructs autophagy-induced apoptosis by sponging miR-5047 and upregulating VEGFA [46]. Considering that 80% of patients with cervical cancer suffer from invasive and metastatic tumors by the time of diagnosis, circ_0023404 could become a valuable biomarker. circ_001971 (gene of origin: *UXS1*) also increases VEGFA expression and enhances invasion and lymphatic metastasis of colorectal cancer [47]. This circRNA also correlated with advanced tumor, node, metastasis (TNM) stage, further establishing VEGFA as a crucial proangiogenic factor. For these reasons, the VEGF pathway has been the target of anticancer treatment (anti-VEGF targeted therapy) for solid tumors. Several promising agents have been developed so far, such as bevacizumab [43], though circRNAs have not been investigated in this context yet. In the following years, the therapeutic potential of circRNAs as vehicles for targeted drug delivery is expected to be examined in detail.

### 4.2. WNT/β-Catenin

The WNT signaling pathway is divided into the β-catenin-dependent pathway, also known as “canonical”, and the β-catenin-independent pathways, which are further divided into the WNT/JNK and WNT/Ca^2+^ pathways. So far, the canonical pathway has been better understood and characterized. In the absence of WNT ligands, cytoplasmic β-catenin (CTNNB1) is targeted and phosphorylated by a destruction complex comprising AXIN2, CSNK1A1, APC, and GSK3B. This leads to the degradation of β-catenin through the 26S proteasome pathway, aided by an E3 ubiquitin-protein ligase. When WNT ligands are present, they bind to Frizzled (FZD) receptors, which are then brought to complex with the lipoprotein receptor-related proteins 5 and 6 (LRP5 and LRP6). Upon activation of the receptor complex, a signal is sent to the phosphoprotein Dishevelled (DVL1). Subsequently, the destruction complex binds to the LRP5 and LPR6 co-receptors and is not able to phosphorylate β-catenin. This results in both the accumulation of β-catenin in the cytoplasm and its transfer to the nucleus, where it interacts with the TCF/LEF transcription factors and promotes the transcription of downstream targets, such as *CCND1*, *MYC*, and *MMP1* [48]. Several reports support that the canonical WNT/β-catenin signaling pathway modulates cancer cell apoptosis, proliferation, invasion, and migration in the context of the initiation and progression of various cancers. Over the past few years, various studies have indicated that circRNAs interact with signaling molecules to influence signal-dependent cell functions and regulate cancer development and progression via the WNT/β-catenin signaling pathway [49]. Intriguing examples of such interactions are discussed below, while a more extensive list is presented in Table 1.

The impact of WNT signaling on the carcinogenesis of colorectal cancer is investigated in several studies which highlight the loss of *APC*. APC is a critical component of the β-catenin destruction complex and an inhibitor of WNT signaling. Besides APC, APC2 also regulates the WNT/β-catenin signaling pathway, and its deregulation is associated with tumorigenesis. Geng et al. noted that circ_009361 (gene of origin: *GNB1*) increased *APC2* expression and inactivated the WNT/β-catenin cascade by sponging miR-582-3p. The circ_009361/miR-582-3p/*APC2* axis is further validated by the fact that in the absence of circ_009361, miR-582-3p targets the mRNA *APC2* and inhibits its expression [59]. Interestingly, the targeting of *APC2* and *APC* by circRNAs has also been observed in osteosarcoma and gastric cancer, with a similar impact on cancer cell growth [52,61].

Evidently, the regulation of this pathway is achieved in several levels. FZD4 is a vital promotive regulator in the activation of the WNT/β-catenin signaling pathway and it is regulated by miR-516b-5p. A recent study designated circ_100290 as a miR-516b-5p sponge. The overexpression of circ_100290 leads to higher protein levels of FZD4 and in this way, it activates the WNT/β-catenin pathway, promotes colorectal cancer cell proliferation, invasion, and migration, and suppresses apoptosis [58]. Another critical inhibitor of WNT signaling is DKK1, which isolates LRP6 so that it cannot aid in the activation of the pathway. circ_0006427 (gene of origin: *BCAR3*) suppresses lung cancer cell invasion, proliferation, and migration by sponging miR-6783-3p, thus enhancing the levels of the miR-6783-3p downstream target, *DKK1*, resulting in the inactivation of the WNT/β-catenin signaling pathway [65].

Moreover, an intriguing study showed that circ-CTNNB1, a circRNA that derives from β-catenin, is overexpressed in malignant states. It is predominantly localized in the nucleus and interacts with the transcriptional regulator DDX3X. This strengthens the interaction between DDX3X and the transcription factor YY1, leading to the *trans*-activation of the latter and an increase in the expression of its downstream genes, such as *WNT1*, *WNT3*, *AXIN*2, and *β-catenin* [68]. Therefore, the circ-CTNNB1/DDX3X/YY1 complex contributes to the activation of the WNT/β-catenin cascade and promotes cancer growth and invasion. In most cases, circRNAs modulate the expression levels of downstream molecules; interestingly, in this study, circ-CTNNB1 did not alter the protein levels of DDX3X, but rather it enhanced the effect of DDX3X on YY1. This alternative mechanism suggests that circ-CTNNB1 can act as a mediator of the WNT/β-catenin signaling pathway without changing the expression levels of its direct target [69].

Additionally, β-catenin is upregulated by circ-ZFR in hepatocellular cancer, through the sponging of miR-3619-5p. This ceRNA function of circ-ZFR leads to the activation of the pathway and the promotion of cancer progression [63]. Interestingly, the binding of β-catenin to the TCF/LEF1 transcription factor can be promoted or inhibited by other proteins and one such protein example with an inhibitory role is CTNNBIP1. However, it was shown that this interaction can be obstructed in papillary thyroid cancer by circ_102171 (gene of origin: *SMURF2*). This circRNA interacts with CTNNBIP1 and consequently blocks its interaction with β-catenin, leading to the activation of the WNT/β-catenin pathway [53]. Another member of the Armadillo catenin family is CTNND1. In contrast to β-catenin, CTNND1 is less investigated, though there are recent indications that it is implicated in oncogenesis. Therefore, its regulation by miR-197-3p, which is sponged by circ_0002577 (gene of origin: *WDR26*) in endometrial carcinoma is an interesting finding, providing new insights regarding the regulatory axis and role of CTNND1. The high levels of CTNND1 were associated with enhanced WNT/β-catenin signaling activity, cancer cell proliferation and migration [56].

One of the most well-investigated circRNAs is circ-ITCH. In the context of the WNT/β-catenin signaling pathway regulation, two independent studies have investigated its role in papillary thyroid cancer and breast cancer and proposed two distinct modes of action for this onco-suppressive circRNA. In papillary thyroid cancer, circ-ITCH sponges miR-22-3p, leading to higher expression rates of its direct target, *CBL*. CBL is an E3 ubiquitin-protein ligase, which targets the active β-catenin and triggers its degradation, resulting in the inactivation of the WNT/β-catenin pathway [54]. In breast cancer, a more complicated mode of action is proposed, via which circ-ITCH increases the expression levels of its linear transcript. More specifically, miR-214-3p and miR-17-5p were sponged by circ-ITCH. Therefore, they were unable to bind to and suppress the expression of their target *ITCH* [55]. ITCH is another E3 ubiquitin-protein ligase, which assists in the degradation of the DVL1 protein and inactivates the WNT/β-catenin pathway, and ultimately induces breast cancer progression. This mechanism of action further implicates *ITCH* and its RNA products in the regulation of WNT/β-catenin signaling, in general.

Finally, an interesting study revealed that circRNAs can regulate the WNT/β-catenin signaling pathway via epigenetic alternations and chromatin remodeling. More specifically, it was shown that circ-MYO10 is upregulated in osteosarcoma and acts as a sponge for miR-370-3p. The sponging of miR-370-3p hinders the binding of this miRNA to its target *RUVBL1*. The latter was found to form a chromatin remodeling complex with the histone-modifying factor KAT5, in order to enhance the acetylation of histone H4K16 in the vicinity of the promoter region of *MYC*, leading to increased expression levels of the latter. Moreover, *RUVBL1* interacts with β-catenin/LEF1 complex. Consequently, these interactions led to increased transcriptional activity of the β-catenin/LEF1 complex, increased activity of the pathway, and enhanced cancer cell proliferation and EMT (Figure 3). These findings designate circ-MYO10 as a potential therapeutic target for this malignancy [51].

All these findings highlight the multifaceted role of circRNAs in the regulation of this pathway and, subsequently, in cancer progression. Given that neither the β-catenin-independent pathway nor its potential regulation via circRNAs have been investigated in the context of tumor progression, it would be fruitful to decipher this potential relation, since it could expand our knowledge regarding cancer pathobiology.

### 4.3. MAPK

The mitogen-activated protein kinases (MAPKs) are members of the large family of Ser/Thr kinases, triggering multiple rounds of hierarchical phosphorylation-activating kinase circles, from the cell surface to the nucleus. They control fundamental cellular processes, such as growth, proliferation, differentiation, migration, and apoptosis. The MAPK signaling cascade is activated following the interaction of one or more growth factors with their specific receptors [receptor tyrosine kinase (RTK) family], integrins, and ion channels. This binding activates the transmembrane receptors and, subsequently, the signal transduction cascade through cytosolic intermediates, and finally leads to the regulation of transcription/translation of effector genes [70]. Four MAPK cascades have been defined based on the components in the MAPK layer: MAPK1 and MAPK3 (ERK subfamily), MAPK8, MAPK9, and MAPK10 (JNK subfamily), MAPK14 (p38), and MAPK7 (ERK5). JNKs are implicated in the WNT signaling pathway independently of β-catenin, as well [48].

Deregulated MAPK signaling is implicated in the progression of a wide range of cancers and occurs via multiple mechanisms, including abnormal expression of RTKs and/or genetic mutations that trigger the activation of RTKs and, thus, of downstream signaling molecules in the absence of appropriate stimuli [71]. Numerous studies focus on the dual role of MAPK signaling pathways that can mediate cell transformation, as well as apoptosis, such as the ERK and JNK signaling pathways [72,73,74]. However, there is still much to be deciphered in terms of regulation and mode of action in both preclinical and clinical research. 

circRNA-driven MAPK signaling regulation is a hot topic of research, and there are several examples of such interactions in a variety of human malignancies. One such interesting example is the circ-ITGA7-mediated MAPK signaling regulation. This circRNA is downregulated in colorectal cancer, while its upregulation is associated with reduced tumor growth. circ-ITGA7 hinders tumor progression in a dual way, in which the expression of its linear transcript is implicated. Specifically, it sponges miR-370-3p, which leads to an increase in the levels of its target, *NF1*. The well-known tumor suppressor NF1 then promotes the hydrolysis of GTP by RAS GTPase, inactivates RAS and negatively regulates the activity of RAS downstream targets, including the RAF/MEK/ERK signaling pathway. The suppressed RAS signaling fails to activate RREB1, a transcription factor that has been associated with tumor development. This leads to elevated expression of the linear transcript of *ITGA7*, which further suppresses tumor growth and metastasis, as shown in Figure 4 [75]. Another circRNA implicated in colorectal cancer progression via RAS signaling regulation is circ_102209, although in this case, the circRNA is oncogenic. More precisely, it acts through the miR-761/*RIN1* axis. RIN1 can positively regulate RAS signaling at different levels. Therefore, its upregulation induces the survival and proliferation of cancer cells [76].

In general, circRNAs can affect the MAPK signaling activity at multiple levels, affecting the binding of the ligand to the receptor, the MAPKs themselves, and the regulators of MAPKs as well. For instance, circRNAs can regulate MAPK signaling by affecting the levels of the IGFBP family. These proteins bind to the insulin-like growth factors (IGFs), prolong their half-life, and regulate their tissue distribution. Additionally, they play a key role in normal cell function and are rarely mutated in cancer. However, studies support that IGFBPs can exert either an oncogenic or an onco-suppressive role, via the regulation of the IGFR/MAPK and IGFR/PI3K/AKT signaling pathways [77]. It has been observed that circRNAs increase the levels of specific oncogenic IGFBPs, and, subsequently, promote cancer progression [78,79]. However, whether cancer progression is mediated through IGFR/MAPK and/or IGFR/PI3K/AKT signaling activation remains to be revealed.

Moreover, circ-AGFG1 upregulates the RAF1 kinase levels, through the sponging of miR-370-3p, thereby initiating the MAPK signaling cascade and enhancing cervical cancer progression [80]. circ-DLST exerts a similar role in gastric cancer, via sponging miR-502-5p and subsequently upregulating its downstream target, *NRAS* [81]. Regarding MAPK signaling regulators, it was shown that circ-ITCH is downregulated in ovarian cancer and exerts its onco-suppressive role through the miR-145-5p/*RASA1* axis; RASA1 is an inhibitor of MAPK signaling. In contrast, NEK2 is an activator that interacts with MAPK1, promotes cell cycle progression, and hence cell proliferation. *NEK2* expression was found to be upregulated by circ-PDSS1 in gastric cancer, which stimulated cancer progression [82].

Intriguingly, circRNAs can promote cancer progression by inactivating MAPK signaling as well. A recent study in non-small cell lung cancer revealed that circ-ZKSCAN1 assisted in the upregulation of *FAM83A* by acting as a ceRNA for miR-330-5p. The high levels of FAM83A, which led to reduced activation of the MAPK signaling pathway and increased cancer cell proliferation [83], is a finding that supports the dual role of this pathway in tumorigenesis. FAM83A belongs to a well-known oncogenic family, which has been implicated with the activation of MAPK signaling in previous studies [84]. Therefore, these results that associate the increased levels of FAM83A with decreased MAPK signaling activity could provide novel insights into its regulatory role. However, further investigation is necessary.

All the aforementioned circRNA interactions are the most intriguing cases out of the several so far investigated, and firmly support the key regulatory role of these novel molecules in the regulation of the MAPK signaling pathway with regard to cancer progression. Although this field is still in its infancy, the complex regulatory networks discussed above provide a new perspective in the matter of metastatic burden. A more detailed list of circRNAs with a regulatory role in MAPK signaling can be found in Table 2.

### 4.4. PI3K/AKT

The PI3K/AKT is an important signaling pathway that is responsible for a variety of cellular activities, such as cell growth, proliferation, differentiation, and migration; this pathway is activated by both EGFR and IGFR signaling. PI3Ks are a family of related intracellular signal transducer enzymes that are activated by the stimulation of RTKs and/or activation of RAS family proteins, while AKT1, a Ser/Thr protein kinase, mediates the PI3K effects on tumor growth and progression. The phosphorylation of AKT1 is associated with cell proliferation and inhibition of apoptosis in several malignancies, since it regulates downstream targets, which promote angiogenesis and cancer progression. The PTEN protein, a known tumor-suppressor that is frequently mutated in cancer, attenuates the activation of the PI3K/AKT pathway. Overall, the PI3K/AKT signaling pathway serves an oncogenic role in the initiation and progression of cancer, and several reports are evidencing that targeted inhibition of this pathway obstructs tumor development [85].

Various studies, prompted by the key role of this pathway, highlight the implication of circRNAs in PI3K/AKT signaling regulation (Table 3), while researchers’ interests have focused on potential intermolecular interactions in malignant states. An interesting example of an oncogenic circRNA is circ-PARD3. This circRNA is overexpressed in laryngeal squamous cell carcinoma and has been linked to poor overall survival and enhanced chemoresistance of cancer patients. It was shown that circ-PARD3 sponges the onco-suppressive miR-145-5p, therefore leading to enhanced expression of its target, *PRKCI*. Subsequently, the kinase PRKCI phosphorylates AKT1, which in turn activates the signaling cascade that decreases autophagy and increases the proliferation rate and migration of cancer cells [96]. Considering the high incidence and mortality rate of this malignancy, further investigation of this circRNA as a potential therapeutic target could prove fruitful.

In general, there are circRNAs that influence the PI3K/AKT signaling pathway directly, by affecting the expression of key components of the pathway, such as by sequestering a miRNA that normally binds to *AKT1* [97], or in a more indirect way, such as by modulating the expression of more distant regulators. Indicative examples are the regulation of MDM2 levels, which is involved in the degradation of TP53, by both circ-FAM53B [98] and circ_0102049 (gene of origin: *ATL1*) in ovarian cancer and osteosarcoma, respectively [99]. The regulation of metastasis-associated proteins by circRNAs constitutes another interesting example of such regulatory interactions; for instance, MTA1 is regulated by circ_0039411 (gene of origin: *MMP2*) in papillary thyroid carcinoma [100] and MET by circ-BFAR in pancreatic cancer [101].

Besides the well-known regulators of the PI3K/AKT signaling pathway, HEMGN has been shown to accelerate thyroid cancer cell proliferation rate via the activation of this pathway. HEMGN can facilitate the proliferation and differentiation of hematopoietic cells via the NF-kB, a downstream target of AKT1. Of note, a recent study revealed an alternative way of HEMGN regulation in papillary thyroid carcinoma, where circ-PSD3 sponge miR-637 increases the levels of HEMGN, and activates PI3K/AKT signaling [106].

It is important to mention that there are circRNAs—for instance, circ_0002577 and circ-RUNX1—which increase the expression levels of *IGF1R* and *IGF1*, respectively, leading to enhanced PI3K/AKT signaling activity, elevated cancer cell proliferation rate, and EMT [110,123]. However, due to the crosstalk between PI3K/AKT and MAPK signaling, the hypothesis of a synergistic function cannot be excluded. In contrast, several circRNAs have an opposite effect on the pathway, such as circ-VRK1 that exerts an onco-suppressive role in esophageal squamous cell carcinoma. It prevents miR-624-3p from binding to the 3′ UTR of *PTEN*. Therefore, circ-VRK1 restores the expression levels of *PTEN*. Consequently, the activity of the PI3K/AKT signaling pathway is obstructed, and tumor progression is hampered as well. Additionally, it was shown that the overexpression of circ-VRK1 could reverse the patients’ radio-resistance, indicating that the targeting of this circRNA can serve as a potential therapeutic approach [112].

Besides solid tumors, the regulatory role of circRNAs in the PI3K/AKT signaling pathway has been investigated in hematopoietic malignancies as well. More specifically, circ_0007841 (gene of origin: *SEC61A1*) is upregulated in plasma samples of patients with multiple myeloma and exerts an oncogenic role via the miR-338-3p/*BRD4* axis [122]. BRD4 is a well-studied member of the human BET (bromodomain and extra terminal) family, which is involved in mitosis and regulation of transcription, and is also over-expressed in various cancers. Moreover, recent studies have designated BRD4 as a novel onco-target protein, possibly due to its impact on the progression of the cell cycle. It is also required for the expression of several key oncogenes, including *MYC* and *BCL2*. In the same study, it was shown that activation of the PI3K/AKT signaling pathway could be regulated through the aforementioned cascade. Additionally, it was uncovered that exosomes generated from mesenchymal stromal cells (MSCs) could accelerate the progression of multiple myeloma via circ_0007841, but the exact mechanism of action remains unknown [122]. Given the critical role of MSCs in multiple myeloma progression and the high regulatory potential of circRNAs, these findings are quite significant and promising.

### 4.5. JAK/STAT

The JAK/STAT signaling pathway constitutes one of the most direct cascades of transmitting information from extracellular signals to the nucleus. Without the need for secondary messenger molecules, the activation of the pathway results in a transcriptional response that ultimately regulates cellular immunity, division rate, cell differentiation, and apoptosis [124]. The pathway consists of JAK binding receptors, intracellular JAKs that transduce cytokine-mediated signals, and STAT proteins, which are transcription factors. When activated, STATs translocate to the nucleus to mediate the activation of target genes, such as *BCL2* and *MYC*. This cascade, when excessively active, can result in tumor formation and cancer progression, as evidenced by various studies [125,126].

The effect of circRNAs on the JAK/STAT signaling pathway has been investigated in several instances so far, in the context of tumor progression (Table 4). Firstly, two recent studies have elucidated the oncogenic role of circRNAs in hepatocellular carcinoma, and interestingly, both have evidenced a positive feedback loop involved in each regulatory axis, as shown in Figure 5 [127,128]. More specifically, circ-LRIG3 expression is linked with aggressive tumor features, including increased tumor size, advanced TNM stage, and vascular invasion [127]. The most intriguing finding of this study was that circ-LRIG3 is predominantly localized in the nucleus and acts as a protein scaffold for EZH2 and STAT3. The subsequent formation of this temporary complex induces the activation of STAT3, which then binds to the promoter of circ-LRIG3, thus regulating its expression in a positive feedback way. The progression of hepatocellular carcinoma is mediated by circ-SOD2 as well, through the induction of epigenetic alterations [128]. In particular, circ-SOD2 sequesters miR-502-5p and drives the restoration of *DNMT3A* expression levels, which is a direct target of this miRNA. DNMT3A is a DNA methyltransferase that increases the methylation of the *SOCS3* promoter and consequently reduces its expression levels. This way, the inhibitory effect of SOCS3 on JAK2 is annulled, and the subsequent activation of JAK2/STAT3 signaling enhances cancer cell growth and migration; interestingly, activated STAT3 binds to the promoter of circ-SOD2 and further promotes its expression through a positive feedback loop. In contrast, circ_0004913 (gene of origin: *TEX2*) was found to have an opposite effect on hepatocellular cancer [129]. It impedes invasion, EMT, and glycolysis by acting as a ceRNA for miR-184, therefore restoring the suppressing role of HAMP on JAK2/STAT3 signaling, and thus, annulling its inhibitory effect on JAK2. Enhanced cancer cell growth and migration are regulated by the activation of JAK2/STAT3 signaling and activated STAT3 further promotes circ-SOD2 expression through a positive feedback loop. In contrast, circ_0004913 (gene of origin: *TEX2*) has an opposite effect on hepatocellular cancer [129]. It impedes invasion, EMT, and glycolysis by acting as a ceRNA for miR-184 and preventing it from binding to *HAMP*, therefore restoring the suppressing role of HAMP on JAK2/STAT3 signaling.

Another feedback interaction was also reported in hepatoblastoma [130]. In particular, circ-STAT3 is transcriptionally activated by GLI2, and at the same time, GLI2 and STAT3 are both upregulated due to circ-STAT3 binding miR-29-3p isomiRs. This regulatory cascade promotes hepatoblastoma cell growth, stemness, and migration. Of note, both miR-29a-3p and miR-29c-3p were found to bind to the 3′ UTR of *VEGFA* as well, further supporting the intricate crosstalk between these pathways [43,47]. Moreover, circ-ZNF124 is implicated in the progression of non-small cell lung cancer, one of the most aggressive types of cancer with the majority of diagnosis at advanced stages [81]. circ-ZNF124 negates the miR-337-3p-mediated repression of JAK2/STAT3, and the consequent activation of this pathway increases cancer cell growth, migration, and colony formation abilities. These malignant features are enhanced by the activation of downstream target genes of STAT3, such as *BCL2* and *FOS*.

**Figure 5 cancers-13-02744-f005:**
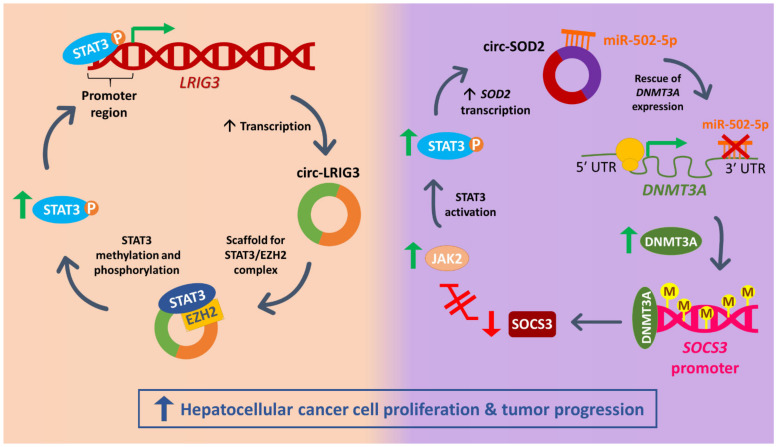
Two examples of a positive feedback regulation of circRNAs expression, as seen in hepatocellular cancer. circ-LRIG3 and circ-SOD2 promote tumor progression through the activation of JAK/STAT signaling. More specifically, circ-LRIG3 acts as a protein scaffold, assisting the methylation of STAT3 by EZH2 and its subsequent phosphorylation, while circ-SOD2 acts as a competitive endogenous RNA and upregulates DNMT3A, which in turn suppresses the inhibitory role of SOCS3 in STAT3 activation. In both cases, activated STAT3 binds to the respective gene promoters and enhances their transcription, thus forming a positive feedback loop. Green arrows denote induction of expression, whereas the red “reverse tau” symbol (⊥) indicates attenuation of expression. Vertical green and red arrows indicate increased or decreased expression levels, respectively. Black arrows signify increased transcription rates, while dark blue arrows indicate the sequence of events in the feedback loops.

The biomarker potential of circRNAs was further supported in gastric cancer, where circ-CUL3 was found to accelerate the Warburg effect progression and impact the overall survival of patients. The Warburg effect, one of the hallmarks of cancer, is an alternative form of cellular metabolism where glucose is mainly converted to lactate [132]. Considering that tumor cells mediate the reprogramming of energy metabolism in order to sustain development, the stimulation of glycolysis and lactate production through the circ-CUL3/miR-515-5p/STAT3/HK2 axis is an intriguing finding. The relation of circ-CUL3, and various other circRNAs, with advanced clinical stage and unfavorable prognosis renders them promising prognostic biomarkers that could be introduced into clinical practice [44,45,132].

### 4.6. TGF-β/SMAD

The transforming growth factor-beta family, mainly known as TGF-β, consists of proteins encoded by three genes, namely *TGFB1, TGFB2*, and *TGFB3*. In particular, they encode three structurally and functionally similar isoforms (TGFB1, TGFB2, and TGFB3); most frequently, each isoform is simply mentioned as TGF-β. TGF-β ligands bind to type II receptors, which act as Ser/Thr kinases, and recruit and phosphorylate type I receptors, thus forming heteromeric complexes. TGF-β stimulates signaling cascades that are responsible for vital cellular procedures, such as morphogenesis, immune regulation, and differentiation. The main mediators of TGF-β signaling are SMAD transcription factors, which regulate the expression of several genes. The role of this pathway in cancer is rather ambiguous, as its function in early tumor development is suppressive, but later on, the secretion of TGF-β facilitates tumor invasion and metastasis [143,144]. Although this pathway has been characterized as simple, it is strictly orchestrated; several circRNAs are reported to be implicated in these regulatory axes in cancerous situations, mainly by acting as miRNA sponges and affecting the expression levels of downstream components that are vital for the efficient operation of the pathway [137,145,146].

Most circRNAs affecting the TGF-β/SMAD signaling pathway have been shown to regulate EMT. To begin with, an important regulator of the TGF-β/SMAD signaling pathway is the transcriptional repressor TRIM33 (T1F1γ), which acts suppressively on this pathway. Specifically, it represses the TGF-β-dependent EMT in mammary epithelial cells, as well as in non-small cell lung cancer cells. *TRIM33* constitutes a direct target of both miR-429 and miR-200b-3p. In non-small cell lung cancer, high expression levels of circ_0008305 (gene of origin: *PTK2*) have been reported; this circRNA sponges both these miRNAs and leads to *TRIM33* upregulation and subsequent repression of the TGF-β-dependent EMT [139]. In contrast, circ_0007294 (gene of origin: *ANKS1B*) promotes EMT and metastasis of triple-negative breast cancer cells by enhancing TGF-β expression. Mechanistically, this circRNA acts as a ceRNA, similarly to circ_0008305. Specifically, it sponges miR-148a-3p and miR-152-3p [136]. Both these miRNAs bind to the mRNA of the USF1 transcription factor. USF1 directly binds to the *TGF-β* promoter and stimulates its transcription [147]. HOXA10, another DNA-binding transcription factor, facilitates *TGFB2* transcription by interacting with elements on its promoter as well [145]. HOXA10 is downregulated by miR-320b; circ-CTDP1, which acts as a ceRNA for miR-320b, upregulates *HOXA10* and, subsequently, promotes nasopharyngeal carcinoma progression through the upregulation of TGFB2 [137].

Besides directly affecting TGF-β, some circRNAs have been witnessed to affect downstream molecules of this pathway to facilitate EMT. A relevant example is that of circ_0072088 (gene of origin: *ZFR*). Its binding to miR-377-3p in bladder cancer cells leads to elevated expression of *ZEB2*. ZEB2 is implicated in the repression of E-cadherin (CDH1) and thus promotes TGF-β-dependent EMT [148], as the downregulation of the adhesion molecule E-cadherin is necessary for this process. These data explain the circ_0072088-mediated progression of bladder cancer cells [83]. Intriguingly, another circRNA deriving from *ZFR* has been reported to promote hepatocellular cancer progression, through the upregulation of β-catenin [63]. Within this framework, the FOXC2 transcription factor, which acts synergistically with SMAD proteins [149], is upregulated in osteosarcoma cells upon miR-526b-5p binding to circ-PVT1, resulting in the promotion of tumor progression and metastasis [135]. circRNAs that have been witnessed to affect cancer progression through the TGF-β/SMAD pathway are summarized in Table 4.

### 4.7. Hippo/YAP

The Ser/Thr kinases STK3 and STK4 (known as Hippo in *D. melanogaster*) phosphorylate other kinases, namely LATS1 and LATS2, which in turn activate the transcriptional factors YAP1 and TAZ (TAFAZZIN). Normally, this pathway, commonly known as the Hippo/YAP signaling pathway, inhibits cell proliferation and promotes apoptosis [150]. Inactivation or deregulation of this signaling pathway in cancer leads to translocation of YAP1 and TAZ in the nucleus, where they reprogram cells to acquire stem-like characteristics. Subsequently, they induce uncontrolled cell proliferation, migration, and angiogenesis [150,151].

Most of the investigated circRNAs that are indirectly involved in this pathway contribute to cancer initiation; however, few circRNAs affecting Hippo/YAP signaling are also associated with tumor progression. Firstly, circ_0000140 (gene of origin: *KIAA0907*) increases the expression levels of *LATS2*, through binding miR-31-5p, as this kinase is downregulated by miR-31-5p. In this way, circ_0000140 promotes YAP1 phosphorylation and inhibits its translocation to the nucleus, leading to suppression of cell growth and metastasis of oral squamous cell carcinoma cells [140]. circ-PPP1R12A is another intriguing case, since not the circRNA itself but a protein encoded by that was shown to facilitate the migration, invasion, and proliferation of colon cancer cells. This impact of circ-PPP1R12A protein was mitigated upon treatment with an inhibitor of the Hippo/YAP signaling pathway, suggesting that this protein exerts its onco-suppressing role through the activation of this pathway [39]. These two cases are briefly presented in Table 4.

### 4.8. Notch

The Notch signaling pathway is responsible for neurogenesis, angiogenesis, and generally tissue development, and is also involved in cell survival and proliferation. Four transmembrane NOTCH receptors (NOTCH1, NOTCH2, NOTCH3, NOTCH4) are found in mammals. Upon binding to the respective ligands, the active intracellular domain of NOTCH receptors is translocated to the nucleus, in order to interact with transcription factors. The role of Notch signaling in cancer is ambivalent, as it has been reported to act either onco-suppressively or onco-promotively [152].

Regarding the role of circRNAs in Notch signaling, they either directly affect the receptor and/or its ligands, or alternatively other regulators of the pathway. A characteristic example is that of circ-NFIX, which acts as a sponge for miR-34a-5p, a miRNA known for its tumor-suppressive role. As *NOTCH1* expression is inhibited by this miRNA, high levels of circ-NFIX in glioma cells triggers the upregulation of *NOTCH1* and, in this case, the progression of the disease [142]. Within this context, circ-NSD2 acts in a similar mechanistic way, by sponging miR-199b-5p and enhancing the expression of the jagged canonical Notch ligand 1 (Jag1) in mice; this axis was proved to be implicated in metastasis of colon cancer [141]. Lastly, circ_0008532 (gene of origin: *CBFA2T2*) affects bladder cancer progression by increasing the levels of the mRNA encoded by its parental gene, through miR-155-5p and miR-330-5p sponging. CBFA2T2 (MTGR1) acts as a Notch signaling suppressor and, in this case, Notch-mediated signaling repression leads to the reinforcement of the migratory and invasive capabilities of bladder cancer cells [153]. These data are also presented in Table 4.

## 5. CircRNAs Affecting Multiple Signaling Pathways

Each signaling pathway does not act independently of the others; there is significant crosstalk between them, leading to the formation of a complex signaling network. The aberrant interplay between pathways is an issue that, when it is better explored and understood, can lead to breakthroughs regarding cancer progression. A representative part of this broad network initiates from RTKs; typical RTK members are EGFR, VEGFR, MET, and DDR2. RTKs activate FAK and SRC tyrosine kinases, which are mediators for many signaling pathways, including the PI3K/AKT and MAPK, while FAK also interacts with the WNT/β-catenin pathway [154,155].

Regarding the role of circRNAs in the context of the intricate signaling network, some of them have been reported to directly affect the function of an RTK [43,134]. These circRNAs are implicated in tumor migration, invasion, and metastasis by affecting a key component of the signaling network and, consequently, disturbing the downstream signaling cascades [42,156]. For instance, DDR2 expression is affected by the miRNA-sponging function of circ-LAMP1 in T-lymphoblastic lymphoma cells. The function of this circRNA leads to the inhibition of miR-615-5p binding to the 3′ UTR of *DDR2*, leading to DDR2 upregulation. In this way, the activity of vital downstream signaling pathways is deregulated, including that of MAPK and PI3K/AKT, resulting in disease progression [157]. In this sense, the upregulation of the RTK MET in esophageal cancer, due to the action of circ-LPAR3, which sponges miR-198 and, thus, upregulates its direct target *MET*, results in increased phosphorylation of its downstream targets, namely AKT1 and MAP2K7, facilitating invasion and migration of esophageal cancer cells [158].

In other cases, circRNAs have been evidenced to affect the kinase complex FAK/SRC, which has been characterized as signaling mediator. More specifically, in bladder cancer, two circRNAs have been reported to disturb FAK/SRC-mediated signaling and result in reduced tumor invasion and metastasis. G3BP2 constitutes a member of the RAS GTPase-activating proteins and is involved in tumor invasion by promoting SRC and FAK phosphorylation. Interestingly, miR-1178-3p binds to the 5′ UTR of *G3BP2*, enhancing its translation. circ-FNDC3B acts as a sponge for miR-1178-3p, leading to the downregulation of G3BP2 and, consequently, to the inhibition of the FAK/SRC-mediated signaling [42]. Similarly, circ-PICALM binds miR-1265 and hence upregulates its direct target, *STEAP4*. The respective protein binds to FAK and prevents its auto-phosphorylation at tyrosine 937 (Y937), affecting multiple downstream signaling pathways [156]. These regulatory interactions are illustrated in Figure 6. Intriguingly, a circRNA deriving from the *FAK* gene was shown to affect the TGF-β/SMAD pathway, as described above [139]. This finding highlights an interesting aspect regarding circRNAs deriving from genes that encode major signaling pathway components, which warrants further investigation.

Finally, some circRNAs affect key components of multiple signaling pathways, which, interestingly, interact with each other. For instance, extensive crosstalk has been evidenced between the JAK/STAT and other pathways, such as MAPK and PI3K/AKT, as previously discussed [129], and it is possible that circRNAs are involved in and contribute to these regulatory axes. For example, circ-PGAM1 acts as a sponge for miR-542-3p and promotes the expression of its direct target, *CDC5L*; the respective protein subsequently upregulates the non-receptor tyrosine kinase PEAK1. In turn, PEAK1 phosphorylates MAPK1 and MAPK3, as well as JAK2, leading to ovarian cancer progression [159]. Similarly, another circRNA is implicated in the regulation of two signaling pathways that are well-known to crosstalk. In particular, circ-CCT3 acts as a sponge for miR-613, which directly targets both VEGFA and WNT3, enhancing colon cancer metastasis [160]. These cases indicate a possible key role of circRNAs in the context of the extensive crosstalk of signaling pathways. These cases, as well as additional examples of circRNAs affecting multiple signaling pathways are presented in Table 5.

**Figure 6 cancers-13-02744-f006:**
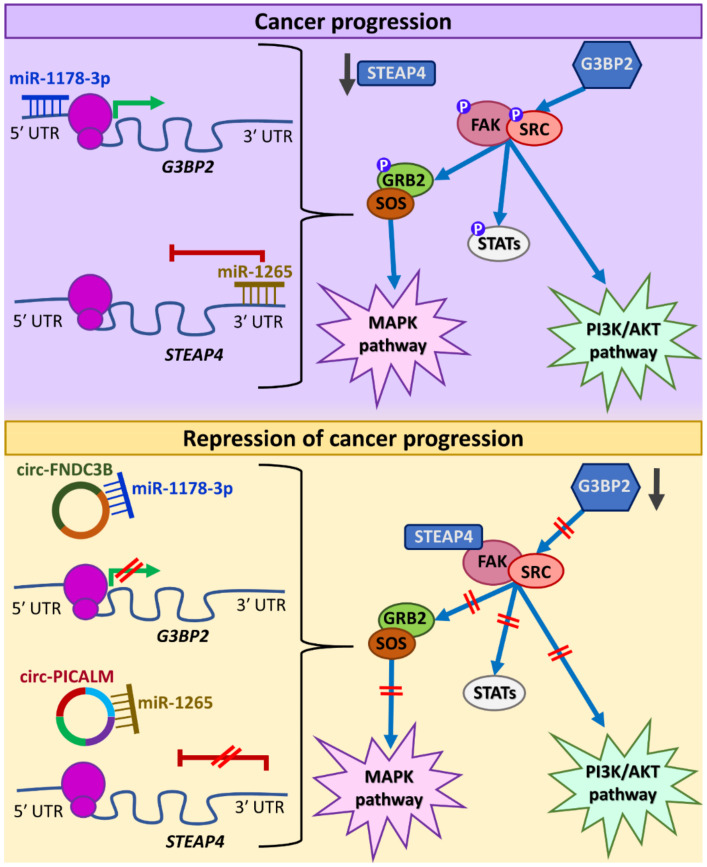
The onco-suppressive role of two circRNAs in bladder cancer. circ-FNDC3B sponges miR-1178-3p, preventing its binding to the 5′ untranslated region (UTR) of *G3BP2,* hence reducing *G3BP2* translation and, consequently, inhibiting the activation of FAK/SRC-mediated signaling. Similarly, circ-PICALM sponges miR-1265 and prevents it from binding to the 3′ UTR of *STEAP4*, this way mediating its upregulation. Thus, STEAP4 binds to FAK and inhibits its phosphorylation, leading to cancer repression. Red “reverse tau” symbols (⊥) indicate attenuation of expression, whereas green arrows denote induction of expression. Blue arrows indicate the sequence of events, while vertical grey arrows indicate decreased expression levels. Τhe parallel red lines (//) indicate the inhibition of an effect.

## 6. Future Perspectives

Considering the multiple biological functions of circRNAs, it is not surprising that numerous researchers have focused on elucidating their distinct properties. For this reason, their implication in the regulation of aberrant signaling, cancer onset, progression, and therapy resistance has repeatedly been demonstrated [16,35], and the most promising areas for future research are shown in Figure 7. The immense potential of highly expressed circRNAs as novel, non-invasive biomarkers has righteously kept them in the spotlight; circulating circRNAs are enriched in body fluids, such as blood, saliva, plasma, and serum, and there are several circRNAs found in exosomes that can affect the tumor microenvironment and malignancy of cancer cells [122]. The conservation, stability, and tissue/cell-specific expression of circRNAs firmly support their great prospect as prognostic and diagnostic liquid biopsy biomarkers. Future studies are expected to elaborate on their sensitivity, specificity, and utility, and bridge the gap between laboratory studies and clinical practice.

The research for novel molecules that can serve as effective cancer therapeutic agents is ongoing, and various approaches have designated circRNAs as promising therapeutic targets. It has been demonstrated that when the activity of oncogenic circRNAs is obstructed, tumor progression is halted and cancer cell apoptosis rate is increased [45]. Furthermore, circRNAs can regulate tumor cell immunity in a variety of ways: they can mediate the activation of innate immunity, assist tumor immune escape through the PD-1/PD-L1 (PDCD1/CD274) checkpoint, inhibit the function of natural killer (NK) cells and promote the NK-mediated responses as well [163]. Besides targeting circRNAs and suppressing or enhancing their function, these molecules could also be utilized as vehicles for targeted drug delivery. circRNAs can effectively carry a therapeutic molecule to its specific target, protecting it from degradation and evading the restrictive role of ABC transporters. Such molecules can have a broad range of targets that include signaling pathway components, for example, they can inhibit angiogenesis by targeting VEGFRs, or induce cell cycle arrest through the WNT/β-catenin pathway. However, research in these treatment approaches is still in its infancy, and their efficacy and safety need to be extensively tested in order to minimize toxic side effects.

**Figure 7 cancers-13-02744-f007:**
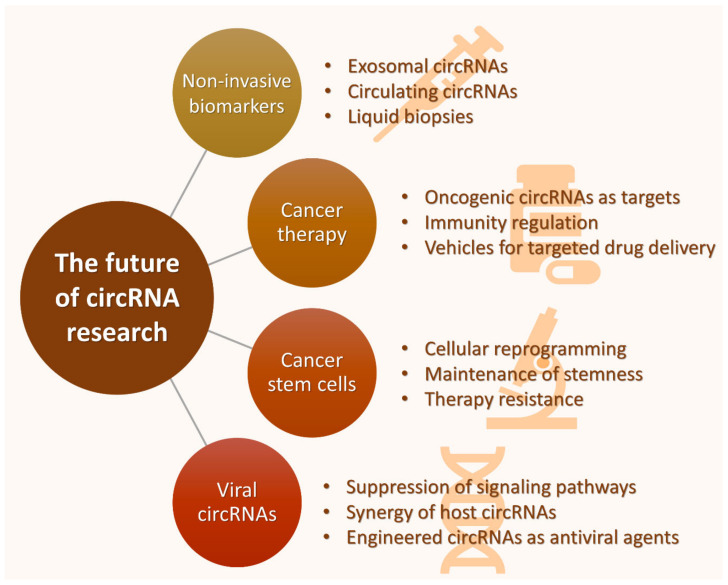
Overview of the most promising areas for circRNA research. circRNAs hold great potential as non-invasive biomarkers for the prognosis, diagnosis, and monitoring of the disease, as targets for anticancer therapy, and vehicles for drug delivery as well. Additionally, circRNAs play a crucial role in the reprogramming of cancer cells to gain and maintain stem-like characteristics, often mediating in treatment failure. Viral circRNAs emerge as intriguing regulators of tumorigenesis and cancer progression that render further investigation, since they restrain the onco-suppressive functions of signaling pathways.

Moreover, an increasing number of studies emphasize the participation of circRNAs in the regulation of signaling pathways linked to cancer stem cells (CSCs) [130]. CSCs are vital for tumor progression, since they promote growth and are responsible for metastatic spread, relapses, and treatment failure. Interestingly, circRNAs can alter the characteristics of cancer cells to gain stem-like properties, such as self-renewal. In addition, some circRNAs can block the PI3K/AKT signaling pathway that supports the maintenance of CSCs, while others favor CSCs maintenance by suppressing the JAK/STAT and Notch signaling pathways. Although the interplay between circRNAs and CSCs has aroused the interest of researchers, the precise mechanisms remain elusive for most cases and only a few circRNAs have been investigated in this field. Considering that circRNAs can indirectly regulate cancer cell stemness, future studies should also focus on upstream molecules that participate in cell differentiation as well, such as EZH2.

Another promising field of circRNA research includes viral circRNAs. Even though circular transcripts were firstly reported in RNA viruses, viral-derived circRNAs have only recently caught the attention of researchers. Several circRNAs are encoded by highly tumorigenic viruses, such as the Epstein–Barr virus, the Kaposi’s sarcoma-associated herpesvirus, the human papillomavirus, and the hepatitis B virus [164]. Moreover, the viral infection affects the expression of host circRNAs that are involved in immunosurveillance and can impair crucial signaling pathways, while it has been reported that host circRNAs may act in groups during an anti-viral response. It is also intriguing that host cells can differentiate between endogenous and exogenous circRNAs, and the immunity against non-self circRNAs renders further exploration since it can potentially be used to counter viral-induced tumorigenesis. However, particular attention should be paid to circRNAs that are produced in the infected cells using the host machinery, as they may be able to surpass sensing for foreign molecules. Additional treatment approaches that need further investigation include the depletion of pro-viral circRNAs, and the development of engineered circRNAs that target specific miRNAs and proteins, which inhibit the function of signaling pathways.

Despite the great progress in research efforts, several limitations remain unsolved. For instance, most studies focus on identifying the downstream targets of a circRNA/miRNA axis, and neglect to assess the availability of the circRNA or the number of binding sites that it possesses; many miRNA sponges are expressed in low copy numbers and are insufficient in exerting a regulatory role. Furthermore, circRNAs are still far from being incorporated into standard clinical practice, and the complicated experimental procedures required for recommending circRNAs with biomarker potential is a major reason for this. Advances in RNA sequencing have allowed breakthroughs in accurate circRNA detection in cancer patients’ samples; however, it is an expensive and time-consuming method, and more studies should explore easier and user-friendlier methods. More large-scale studies are required as well, prior to affirming the utility of tumor-specific circRNAs in cancer diagnosis, prognosis, and treatment. Nonetheless, the research interest for circRNAs is rapidly growing, and our understanding of these molecules and their intricate functions will be more thorough as more studies are conducted and better analysis pipelines are described.

## 7. Conclusions

Overall, circRNAs interact multifariously with major signaling pathways and affect the progression of cancer. They can regulate angiogenesis through the VEGF signaling pathway, cancer cell proliferation through the WNT/β-catenin signaling pathway, tumor growth and invasion through the MAPK signaling pathway, and apoptosis through the Hippo/YAP signaling pathway, to name a few. Evidently, signal transduction is not limited to the function of a single pathway; several pathways intertwine and work in conjunction to drive cancer progression, and circRNAs reinforce or hinder this crosstalk in various ways. Future studies are expected to further establish circRNAs as effective biomarkers to monitor cancer progression, create more opportunities on targeted therapies, and shed light on intriguing circRNA aspects that remain elusive.

## Figures and Tables

**Figure 2 cancers-13-02744-f002:**
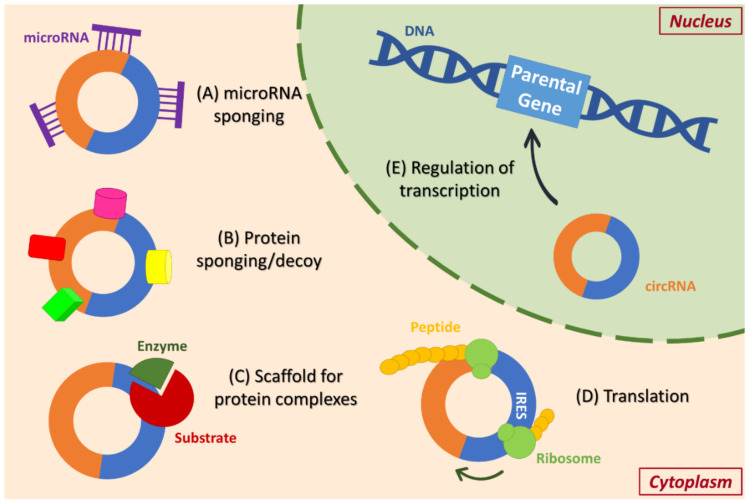
The main functions of circular RNAs (circRNAs). The most frequently observed mechanism of circRNA action is the sponging of microRNA molecules (**A**), thereby preventing them from binding to their mRNA targets. circRNAs restrict the function of proteins as well, by acting as protein sponges/decoys (**B**), which can lead to their retainment and stabilization in a specific cellular compartment. Such proteins are depicted here in various shapes of pink, yellow, green, and red color. The interaction between circRNAs and RNA-binding proteins also facilitates the formation of complexes (**C**) since an enzyme and its substrate(s) are brought to proximity through a circRNA scaffold. Interestingly, several circRNAs can encode peptides in a Cap-independent manner (**D**), and they possess at least one internal ribosome entry site (IRES). Regarding circRNAs that are predominantly located in the nucleus, they usually regulate the transcription rates of their parental gene (**E**).

**Figure 3 cancers-13-02744-f003:**
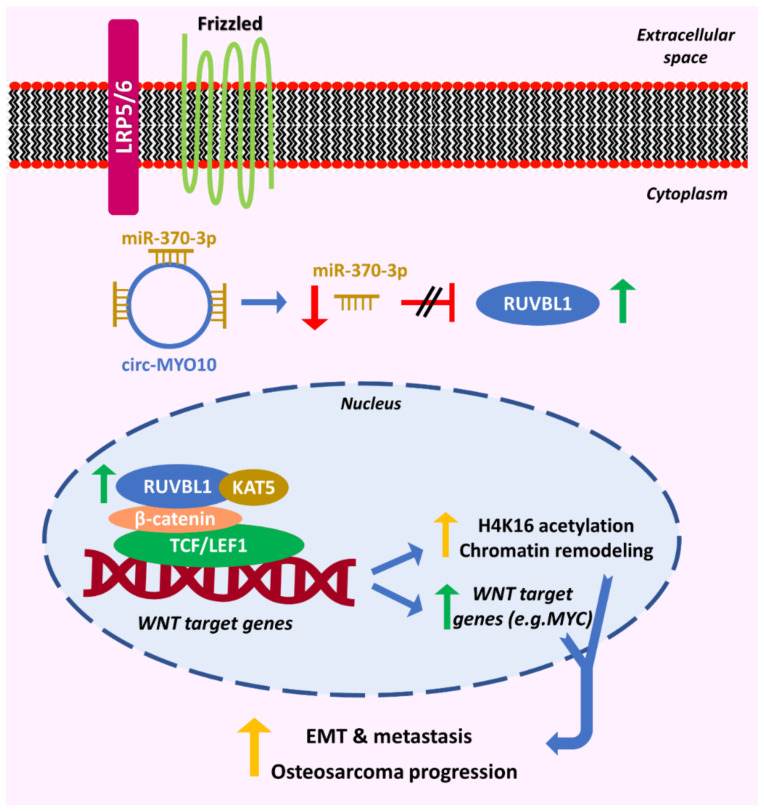
A proposed mode of action of circ-MYO10 regarding the regulation of the WNT/β-catenin pathway in osteosarcoma. circ-MYO10 sponges miR-370-3p, and, therefore, the levels of the latter are not sufficient for reducing the expression of *RUVBL1*. Then, RUVBL1 is transferred to the nucleus, where it interacts with β-catenin and KAT5 and forms a complex that is recruited to the promoter region of *MYC*. This interaction leads to increased acetylation of H4K16 and expression of WNT-targeted genes, including *MYC*. Consequently, enhanced osteosarcoma cell proliferation, epithelial to mesenchymal transition (EMT), and metastasis are attained. The red “reverse tau” symbol (⊥) indicates attenuation of expression, while the parallel black lines (//) indicate the inhibition of an effect. Vertical green and red arrows indicate increased or decreased expression levels, respectively. Vertical yellow arrows denote the increased rate of an event, and blue arrows indicate the sequence of events.

**Figure 4 cancers-13-02744-f004:**
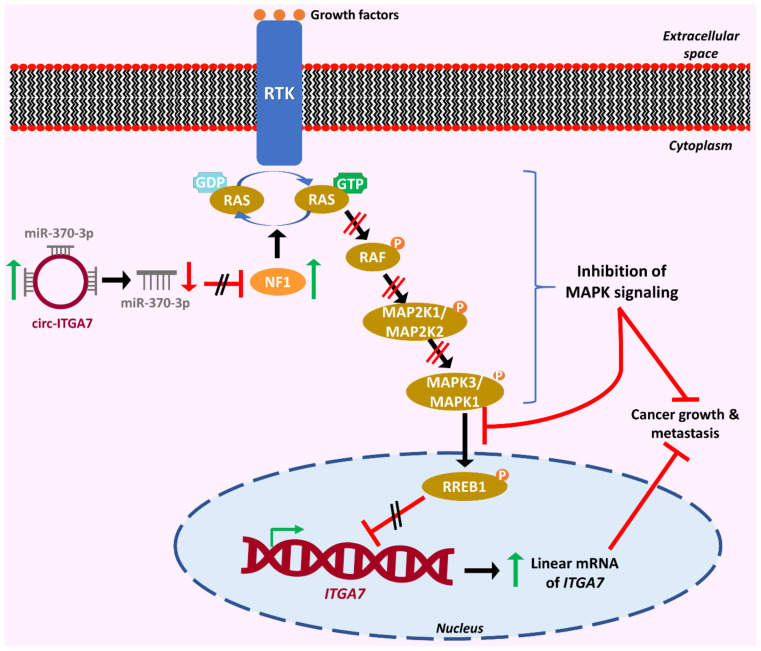
The onco-suppressive role of circ-ITGA7 in colorectal cancer. circ-ITGA7 sponges miR-370-3p, which leads to the increased expression of *NF1*, one of the target genes of miR-370-3p. NF1 promotes the hydrolysis of GTP by RAS GTPase, and therefore inhibits RAS signaling, which consequently obstructs the binding of the transcription factor RREB1 to the promoter of *ITGA7*. This results in enhanced expression of the linear mRNA of *ITGA7*. The inactivation of RAS signaling, accompanied by the increased expression of *ITGA7*, leads to the suppression of cancer progression and metastasis. The red “reverse tau” symbol (⊥) indicates attenuation of expression, while the parallel black and red lines (//) indicate the inhibition of an effect. Vertical red arrows indicate decreased expression levels. Green arrows indicate increased expression levels, and black arrows indicate the sequence of events. The blue arrow indicates the transition from one state to another.

**Table 1 cancers-13-02744-t001:** circRNAs with a regulatory effect in cancer progression through the VEGF and WNT/β-catenin signaling pathways.

Pathway	Cancer	circRNA	Gene of Origin	Interactions	Effect	Reference
VEGF	Bladder cancer	circ-MYLK	*MYLK*	miR-29a-3p/*VEGFA*	Promotes cancer progression	[43]
	Cervical cancer	circ_0023404	*RNF121*	miR-5047/*VEGFA*	Promotes cancer metastasis and chemoresistance	[46]
	Colorectal cancer	circ_001971 (circ_0001060) ^1^	*UXS1*	miR-29c-3p/*VEGFA*	Promotes cancer growth, angiogenesis, and metastasis	[47]
	Renal cell carcinoma	circ-MYLK	*MYLK*	miR-513a-5p/*VEGFC*	Promotes cancer growth and metastasis	[44]
	Pancreatic cancer	circ-NFIB_1_ (circ_0086375) ^1^	*NFIB*	miR-486-5p/ *PIK3R1*/VEGFC	Inhibits lymphangiogenesis and lymphatic metastasis, suppresses cancer progression	[45]
WNT/β-catenin	Glioblastoma multiforme	circ_0043278	*TADA2A*	miR-638	Promotes cancer progression	[50]
	Osteosarcoma	circ-MYO10	*MYO10*	miR-370-3p/*RUVBL1*	Regulates chromatin remodeling; promotes cancer progression	[51]
		circ_0002052	*PAPPA*	miR-1205/*APC2*	Suppresses cancer progression	[52]
	Papillary thyroid cancer	circ_102171	*SMURF2*	CTNNBIP1	Promotes cancer progression	[53]
		circ-ITCH	*ITCH*	miR-22-3p/*CBL*	Suppresses cancer progression	[54]
	Breast cancer	circ-ITCH	*ITCH*	miR-214-3p/*ITCH*, miR-17-5p/*ITCH*	Suppresses cancer progression	[55]
	Endometrial	circ_0002577	*WDR26*	miR-197-3p/*CTNND1*	Promotes cancer progression	[56]
	Esophageal squamous cell carcinoma	circ-GSK3B	*GSK3B*	-	Promotes metastasis	[57]
	Colorectal cancer	circ_100290	*-*	miR-516b-5p/*FZD4*	Promotes cancer progression	[58]
		circ_0009361	*GNB1*	miR-582-3p/*APC2*	Suppresses cancer progression	[59]
	Gastric cancer	circ-HECTD1	*HECTD1*	miR-1256	Promotes glutaminolysis and cancer progression	[60]
		circ-HIPK3	*HIPK3*	WNT1, TCF4, β-catenin	Promotes cancer progression	[20]
		circ-FGD4	*FGD4*	miR-532-3p/*APC*	Suppresses cancer progression	[61]
		circ-REPS2	*REPS2*	miR-558/*RUNX3*		[62]
	Hepatocellular carcinoma	circ-ZFR	*ZFR*	miR-3619-5p/*β-catenin*	Promotes cancer progression	[63]
		circ_0067934	*PRKCI*	miR-1324/*FZD5*	Promotes tumor growth and metastasis	[64]
	Lung cancer	circ_0006427	*BCAR3*	miR-6783-3p/*DKK1*	Suppresses cancer progression	[65]
	Non-small cell lung cancer	circ_000984 (circ_0001724) ^1^	*CDK6*	-	Promotes cell proliferation and metastasis	[66]
	Retinoblastoma	circ-TET1	*TET1*	miR-492, miR-494-3p	Suppresses cancer progression	[67]

^1^ Names in parentheses denote the circRNA ID or alias, as listed on circBase.

**Table 2 cancers-13-02744-t002:** circRNAs with a regulatory effect in cancer progression through the MAPK signaling pathway.

Cancer	CircRNA	Gene of Origin	Interactions	Effect	Reference
Breast cancer	circ_0006528	*PRELID2*	miR-7-5p	Promotes cancer progression	[86]
	circ-WWC3	*WWC3*	miR-26b-3p, miR-660-3p		[87]
Ovarian	circ-ITCH	*ITCH*	miR-145-5p/*RASA1*	Suppresses cancer progression	[88]
	circ-PIP5K1A	*PIP5K1A*	miR-661/*IGFBP5*	Promotes cancer progression	[78]
	circ_102958	*-*	miR-1205/*SH2D3A*		[89]
Cervical cancer	circ-SMARCA5	*SMARCA5*	miR-432-5p	Promotes cell proliferation	[90]
	circ-AGFG1	*AGFG1*	miR-370-3p/*RAF1*	Promotes cancer progression	[80]
Clear Cell Renal Cell Carcinoma	circ-APBB1IP	*APBB1IP*	-	Promotes cancer progression	[91]
Bladder cancer	circ-CEP128	*CEP128*	miR-145-5p/*MYD88*	Promotes cancer progression	[92]
	circ-VANGL1	*VANGL1*	miR-1184/*IGFBP2*		[79]
Colorectal cancer	circ_102209	*-*	miR-761/*RIN1*	Promotes cell growth and metastasis	[76]
	circITGA7	*ITGA7*	miR-370-3p/*NF1*/ RREB1/ITGA7	Suppresses tumor growth and metastasis	[75]
Gastric cancer	circ_0032821	*CEP128*	-	Promotes cell proliferation and metastasis	[93]
	circ-PDSS1	*PDSS1*	miR-186-5p/*NEK2*	Promotes cancer progression	[82]
	circ-DLST	*DLST*	miR-502-5p/ *NRAS*/MAP2K1/MAPK1,miR-502-5p/ *NRAS*/MAP2K1/MAPK3		[81]
Hepatocellular carcinoma	circ-ASAP1	*ASAP1*	miR-326/*MAPK1*, miR-532-5p/*MAPK1*	Promotes cancer progression	[94]
Non-small cell lung cancer	circ-FOXM1	*FOXM1*	miR-1304-5p/*PPDPF*, miR-1304-5p/*MACC1*	Promotes cancer progression and metastasis	[95]
	circ-ZKSCAN1	*ZKSCAN1*	miR-330-5p/*FAM83A*	Promotes cancer progression	[83]

**Table 3 cancers-13-02744-t003:** circRNAs with a regulatory effect in cancer progression through the PI3K/AKT signaling pathway.

Cancer	circRNA	Gene of Origin	Interactions	Effect	Reference
Glioma	circ_0015758	*CFH*	miR-149-3p/*AKT1*	Promotes cancer progression	[97]
	circ_0037655	*CREBBP*	miR-214-3p/*PI3K*		[102]
	circ_0014359	*NUP210L*	miR-153-3p		[103]
Glioblastoma	circ_0067934	*PRKCI*	-	Promotes cancer progression	[104]
Osteosarcoma	circ_0102049	*ATL1*	miR-1304-5p/*MDM2*	Promotes cancer progression	[99]
Laryngeal squamous cell carcinoma	circ-PARD3	*PARD3*	miR-145-5p/ *PRKCI*/AKT1/MTOR	Promotes cancer progression	[96]
Thyroid Carcinoma	circ_0067934	*PRKCI*	-	Promotes cancer progression	[105]
Papillary thyroid carcinoma	circ-PSD3	*PSD3*	miR-637/*HEMGN*	Promotes cancer progression	[106]
	circ_0039411	*MMP2*	miR-1179/*ABCA9*, miR-1205/*MTA1*	Promotes cancer progression	[100]
Breast cancer	circ_103809		-	Promotes cancer progression	[107]
	circ_001569 (circ_0000677) ^1^	*ABCC1*	-		[108]
Ovarian cancer	circ-FAM53B	*FAM53B*	miR-646/*VAMP2*, miR-647/*MDM2*	Promotes cancer progression	[98]
	circ-RHOBTB3	*RHOBTB3*	-	Suppresses cancer progression	[109]
Endometrial cancer	circ_0002577	*WDR26*	IGF1R/PI3K/AKT	Promotes cancer progression	[110]
Bladder cancer	circ-SLC8A1	*SLC8A1*	miR-130b-3p, miR-494-3p	Suppresses cancer progression	[111]
Esophageal squamous cell carcinoma	circ-VRK1	*VRK1*	miR-624-3p/ *PTEN*/PI3K/AKT	Suppresses cancer progression	[112]
Colorectal cancer	circ-RUNX1	*RUNX1*	miR-145-5p/*IGF1*	Promotes cancer progression	[45]
	circ_0026344	*ACVRL1*	miR-183-5p	Suppresses cancer progression	[113]
Gastric cancer	circ_0047905	*SERPINB5*	miR-4516, miR-1227-5p	Promotes cancer progression	[114]
	circ_0010882	*RPL11*	-		[115]
	circ-NRIP1	*NRIP1*	miR-149-5p		[116]
	circ-PSMC3	*PSMC3*	miR-296-5p	Suppresses cancer progression	[117]
Pancreatic cancer	circ-BFAR	*BFAR*	miR-34b-5p/ *MET*/AKT1	Promotes cancer progression	[101]
Lung cancer	circ-BANP	*BANP*	miR-503-3p/*LARP1*	Promotes cancer progression	[118]
Non-Small Cell Lung Cancer	circ-HIPK3	*HIPK3*	miR-107/*BDNF*	Promotes cancer progression	[119]
	circ-GFRA1	*GFRA1*	miR-188-3p		[120]
Melanoma	circ_0001591 (circ_001436) ^1^	*HIST1H2AG*	miR-431-5p/ *ROCK1*/PI3K/AKT	Promotes cell proliferation and cancer progression	[121]
Multiple myeloma	circ_0007841	*SEC61A1*	miR-338-3p/*BRD4*	Promotes cancer progression and cell proliferation	[122]

^1^ Names in parentheses denote the circRNA ID or alias, as listed on circBase.

**Table 4 cancers-13-02744-t004:** circRNAs affecting cancer progression through the JAK/STAT, TGF-β/SMAD, Hippo/YAP, and Notch signaling pathways.

Pathway	Cancer	CircRNA	Gene of Origin	Interactions	Effect	Reference
JAK/STAT	Glioma	circ-HIPK3 (circ_0000284) ^1^	*HIPK3*	miR-124-3p/*STAT3*	Promotes cancer progression	[131]
	Gastric cancer	circ-CUL3 (circ_0008309) ^1^	*CUL3*	miR-515-5p/ *STAT3*/HK2	Promotes Warburg effect progression	[132]
	Hepatocellular cancer	circ-LRIG3 (circ_0027345) ^1^	*LRIG3*	EZH2/STAT3	Promotes cancer progression	[127]
		circ-SOD2 (circ_0004662) ^1^	*SOD2*	miR-502-5p/*DNMT3A*/ SOCS3/JAK2/STAT3	Induces epigenetic alterations, promotes cancer progression	[128]
		circ_0004913	*TEX2*	miR-184/*HAMP*/ JAK2/STAT3	Suppresses cancer progression	[129]
	Hepatoblastoma	circ-STAT3(circ_0043800) ^1^	*STAT3*	miR-29-3p family/ *STAT3*, miR-29-3p family/*GLI2*	Promotes cancer progression	[130]
	Non-small cell lung cancer	circ-ZNF124	*ZNF124*	miR-337-3p/ *JAK2*/STAT3	Promotes cancer progression	[81]
	Esophageal squamous cell carcinoma	circ_0000654 (circ_000608) ^1^	*CHO2*	miR-149-5p/ *IL6*/STAT3	Promotes cancer progression	[133]
	Pancreatic cancer	CDR1as (ciRS-7, circ_0001946) ^1^	*CDR1*	miR-7-5p/*EGFR*/STAT3	Promotes cancer progression	[134]
TGF-β/SMAD	Osteosarcoma	circ-PVT1	*PVT1*	miR-526b-5p/ *FOXC2*	Promotes metastasis	[135]
	Breast cancer	circ_0007294	*ANKS1B*	miR-148a-3p/*USF1*, miR-152-3p/*USF1*	Promotes metastasis	[136]
	Nasopharyngeal carcinoma	circ-CTDP1	*CTDP1*	miR-320b/*HOXA10*	Promotes metastasis	[137]
	Bladder cancer	circ_0072088	*ZFR*	miR-377-3p/*ZEB2*	Promotes metastasis	[83]
	Gastric cancer	circ- CCDC66	*CCDC66*	-	Promotes cell growth and metastasis	[138]
	Non-small cell lung cancer	circ_0008305	*PTK2 (FAK)*	miR-200b-3p/*TRIM33*, miR-429/*TRIM33*	Suppresses metastasis	[139]
Hippo/ YAP	Colon cancer	circ-PPP1R12A	*PPP1R12A*	-	Promotes migration, invasion, and proliferation	[39]
	Oral squamous cell carcinoma	circ_0000140 (circ_002059) ^1^	*KIAA0907*	miR-31-5p/*LATS2*	Suppresses cell growth and metastasis	[140]
Notch	Colon cancer	circ-NSD2	*NSD2*	miR-199b-5p/*Jag1*	Promotes metastasis (in mice)	[141]
	Bladder cancer	circ_0008532	*CBFA2T2*	miR-155-5p/*CBFA2T2*, miR-330-5p/*CBFA2T2*	Promotes migration and invasion	[47]
	Glioma	circ-NFIX	*NFIX*	miR-34a-5p/*NOTCH1*	Promotes cancer progression	[142]

^1^ Names in parentheses denote the circRNA ID or alias, as listed on circBase.

**Table 5 cancers-13-02744-t005:** circRNAs with a regulatory effect in multiple signaling pathways.

Cancer	circRNA	Gene of Origin	Interactions	Effect	Reference
Ovarian cancer	circ-PGAM1	*PGAM1*	miR-542-3p/*PEA K1*/MAPK1, miR-542-3p/*PEAK1*/MAPK3, miR-542-3p/*PEAK1*/JAK2	Promotes cancer progression	[159]
Esophageal cancer	circ-LPAR3	*LPAR3*	miR-198/*MET*/MAP2K7, miR-198/*MET*/AKT1	Promotes migration and invasion	[158]
Bladder cancer	circ-SEMA5A	*SEMA5A*	miR-330-5p/*ENO1*	Promotes cancer progression	[161]
	circ-FNDC3B	*FNDC3B*	miR-1178-3p/*G3BP2*/ SRC/FAK	Suppresses tumor invasion and metastasis	[42]
	circ-PICALM	*PICALM*	miR-1265/*STEAP4*/FAK	Suppresses tumor invasion and metastasis	[156]
Colon cancer	circ-CCT3	*CCT3*	miR-613/*WNT3*, miR-613/*VEGFA*	Promotes cancer metastasis	[160]
Gastric cancer	circ_0001649 (circ_001599) ^1^	*SHPRH*	miR-20a-5p	Suppresses cancer progression	[162]
T-cell lymphoblastic lymphoma	circ-LAMP1	*LAMP1*	miR-615-5p/*DDR2*	Promotes cancer progression	[157]

^1^ The name in parenthesis denotes the circRNA alias, as listed on circBase.

## Data Availability

No new data were created or analyzed in this study.

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
