# Peer review of "Circular RNAs: Emerging Regulators of the Major Signaling Pathways Involved in Cancer Progression"

_cancers, 2021, doi:10.3390/cancers13112744_

Round 1

Reviewer 1 Report

A well reviewed paper in the field of non-coding rna research. Recommended for acceptance and publication.

Author Response

We sincerely thank the Reviewer for the positive appraisal of our review article.

Reviewer 2 Report

This manuscript submitted by Papatsirou et al, discussed the role of circular RNAs (circRNAs) in cancer progression by regulating the various signaling pathways which are primarily involved in tumor development. The review article has explained every aspect of the topic very carefully and in detail with figures and tables wherever required. Although the manuscript may provide some interesting information, however, there are some minor concerns I have for this manuscript. 

Comments-

  1. Since the review is based on the circular RNAs, it would be good if author can prepare a figure which explain the biogenesis and types of circRNAs in section-2.

  1. In line no. 136-137, author mentioned that “circRNAs were initially considered as non-coding RNAs”, it is not clear whether they are still considered as non-coding RNAs or not and why? As some recent studies in literature still mentioned that they are a type of non-coding RNAs (Kong et al. Molecular Cancer, 2020;19:82). I believe since this is a systematic and comprehensive review, author should explain this point too.

Author Response

 1.      We thank the Reviewer for this suggestion. We added a new figure (Figure 1) that illustrates the biogenesis and types of circRNAs in the section “2.1  Biogenesis of circRNAs”.

2.      Prompted by the Reviewer’s comment, we clarified this issue in the revised manuscript:

Page 4 (lines 153-158): Albeit initially considered as non-coding RNAs having only a miRNA-sponging function, some circRNAs have been shown to contain multiple open reading frames (ORFs) and encode proteins, several of which are involved in major cellular processes [33]. The translation occurs in a Cap-independent manner and several circRNAs possess at least one internal ribosome entry site (IRES), while the process can be promoted by the nucleotide modification N6-methyladenosine (m6A) [33,34].

Reviewer 3 Report

This is an excellent  review on circular RNAs and their roles in regulating signalling pathways involved in cancer. The authors have done a very good job in summarising the literature and providing a comprehensive, in depth and up-to-date review of the topic. The authors made an excellent use of diagrams and tables. 

Author Response

(The authors gave the same response as above.)
